**Comment**

# We need to understand economic inequality from children's perspectives

Eddie Brummelman, Richard E. Ahl, Elisabetta Aurino, Sahba N. Besharati, Benjamin W. Domingue, Catherine Lebel, Julia A. Leonard, Dana C. McCoy, Luca M. Pesando, Samuel S. Urlacher, David S. Yeager, Jason Yip & Katherine McAuliffe

Economic inequality is on the rise, and children will bear its burden. Yet children's perspectives are rarely considered. We urgently need interdisciplinary research to better understand how inequality gets into children's heads and under their skins, and to inform policies that center children's lived experiences.

Today's children are growing up in an age of economic inequality. If people sat on their wealth stacked in $100 bills, most of the world would be sitting on the floor, a middle-class person would sit at chair height, and the richest people would be in outer space[1]. Children will be confronted with this type of inequality as they grow up, and so will their children and their children's children. Yet we lack an understanding of how children perceive and respond to it.

Economic inequality is not the same as socioeconomic status (SES). A vast body of research shows that children from low-SES backgrounds often have poorer mental health, physical health, and academic achievement than children from high-SES backgrounds. However, this work does not address *economic inequality*—that is, the unequal distribution of valuable goods and opportunities within children's classrooms, schools, neighborhoods, states, or countries.

Children actively try to make sense of their environments so that they can form mental models of how to survive and thrive within them. Unequal societies determine the allocation of valuable resources such as income, wealth, opportunities, and even the application of the law, so they are a powerful source of information for children's mental models. Thus, it is not only children's own economic resources that matter, but also how unequal their environments appear to them. In fact, children's experiences of poverty or affluence may depend on their perceptions of inequality—whether they see those around them as having similar levels of wealth or as differing greatly in wealth (Fig. 1).

Much research has examined adults' perceptions of inequality, showing that they are consequential, even if miscalibrated[2]. But children's perspectives are often overlooked. We suggest that a developmental perspective is essential to understand how inequality shapes human development. Childhood experiences of inequality may have lasting consequences. As children are forming mental models of the world, they may readily internalize experiences of inequality, coming to view the world as a place where economic success is reserved for only a select few. Additionally, experiences of inequality may shift substantially across development. In early childhood, before formal schooling, children's lives are often segregated by SES. This segregation tends to be stronger in societies with greater economic inequality—meaning that children may not perceive the full extent of inequality in their environments. School transitions, especially into secondary school, often expose children to greater economic inequality, increasing its salience in their daily lives[3].

What are the consequences of growing up in an environment of economic inequality? Scholars across the social, behavioral, and biological sciences have begun to address this question. They have conducted what we call *individual-level research* (which examines how children perceive and respond to resource inequality in their immediate environments) and *structural-level research* (which examines how broader societal patterns of economic inequality are related to children's developmental outcomes).

Individual-level research seeks to identify the psychological mechanisms that explain how children experience resource inequality in their immediate environments. This work typically involves young children and relies on small-scale laboratory experiments that introduce children to varying levels of resource inequality (e.g., stickers or toys). This research shows that, from infancy, children detect and make sense of resource inequality[4]. From age 5, children categorize peers by SES based on subtle cues like clothing[5]. When confronted with inequality, children often make efforts to rectify it—even when doing so requires sacrificing their own resources. They do so especially when they perceive the inequality as unfair or undeserved[6]. As they move into adolescence, children come to understand the structural sources that contribute to economic inequality.

Structural-level research seeks to provide generalizable and ecologically valid inferences about the consequences of real-world economic inequality for children's development[7]. This work typically links large-scale cross-sectional data on adolescents (e.g., Program for International Student Assessment, Health Behavior in School-aged Children) with country-level indicators of economic inequality (e.g., Gini index). This research has revealed associations between country-level economic inequality and children's mental health, physical health, and social relationships. For example, across 34 countries in North America and Europe, adolescents in more economically unequal societies had worse mental health, especially if they were from low-SES backgrounds[8].

Although both lines of research have begun to decipher how children perceive and respond to economic inequality, they have largely remained siloed. This means that little is known about how children perceive economic inequality in their everyday lives, and how their perceptions translate into their mental health, physical health, and social relationships. We call on social, behavioral, and biological scientists to bridge individual- and structural-level research in addressing this urgent question (Table 1).

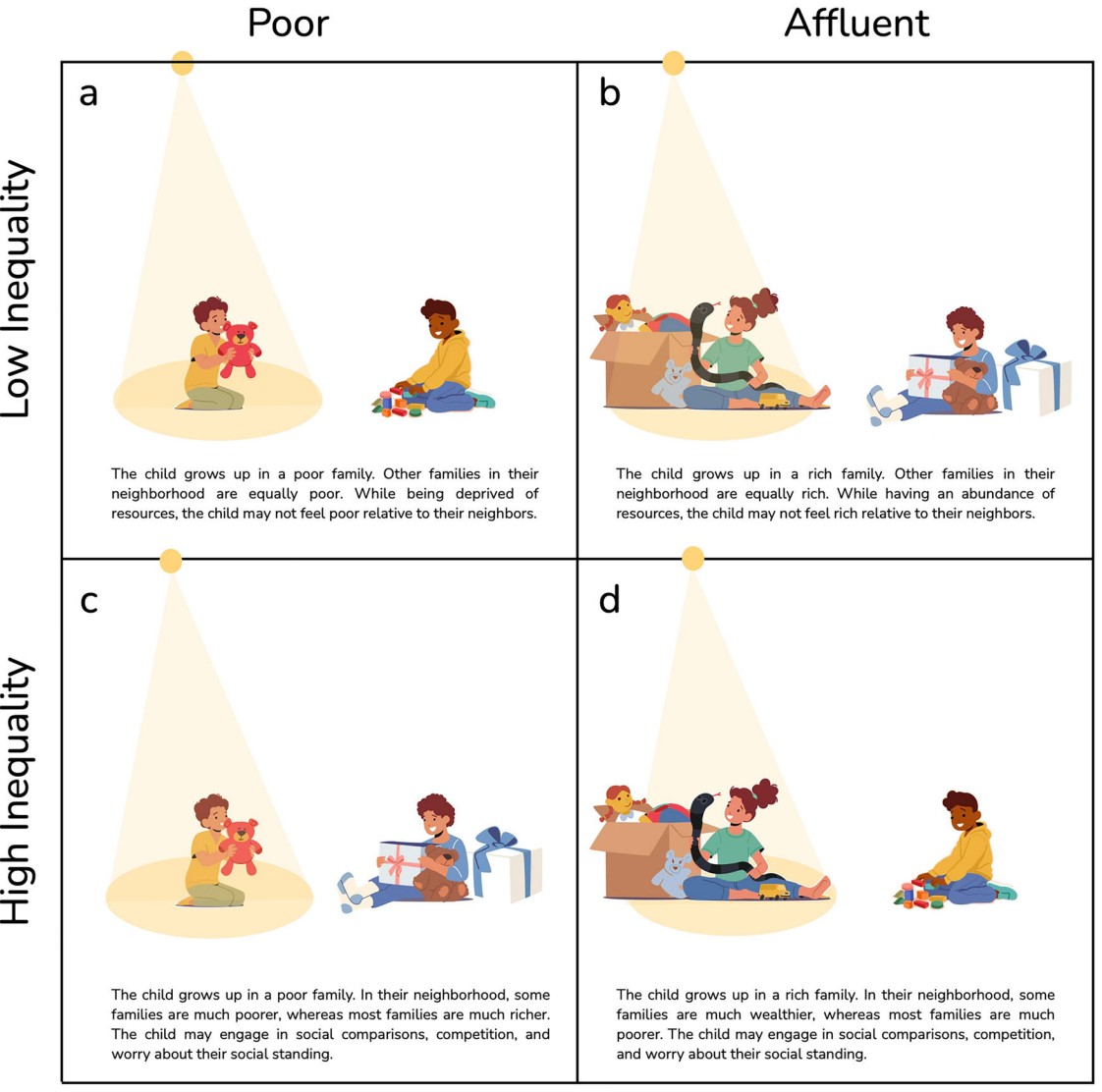

**Fig. 1 | A child's economic resources versus perceived economic inequality.** An illustration of how a child's economic resources and perceived economic inequality can vary independently, creating different sets of psychological experiences. The number of toys denotes the level of resources. Children can perceive inequality from a first-person perspective (i.e., inequality between themselves and another person) and third-person perspective (i.e., inequality between others within a population). To highlight the contrast with children's economic resources (which they perceive from a first-person perspective), the illustration reflects children's first-person perspective on inequality. The spotlight highlights the child of interest. **a** A child living in poverty perceiving low inequality. **b** A child living in affluence perceiving low inequality. **c** A child living in poverty perceiving high inequality. **d** A child living in affluence perceiving high inequality. The figure was created by Katherine McAuliffe using Keynote. The graphic elements were sourced from Freepik under a Premium license and are used in accordance with their terms of use.

How do children's perceptions of economic inequality affect them? Research with adults offers clues. In contexts of economic inequality, adults start to compare themselves with others based on their SES—seeing themselves as either "haves" or "have-nots"—and these labels can take on personal significance. Over time, this can foster a competitive ethos, in which individuals focus on advancing themselves while viewing others as obstacles to success[9]. Those from low-SES backgrounds may experience this as particularly threatening, which may undermine their health and reduce their trust in others and institutions. Because children are often aware of their standing within SES hierarchies[10], they may experience similar downstream consequences of perceived economic inequality.

Children's perceptions of economic inequality not only affect their individual development but also broader societal dynamics, including political extremism. This impact may be strongest in adolescence, when young people often act as agents of social change. For some adolescents, perceptions of inequality can spark collective action and movements advocating for redistribution of wealth (e.g., Occupy Wall Street, Black Lives Matter, Fridays for Future). For others, however, these same perceptions may foster disillusionment, resentment, and, in extreme cases, far-right radicalization or ideologically motivated nationalist violence.

In an attempt to reduce the harms of rising economic inequality, policymakers often promote economic desegregation in schools and

**Table 1 | Open questions addressing children's perspectives on economic inequality**

| | |
|---|---|
| Development | From what age do children spontaneously compare others in terms of socioeconomic status? When do children start to perceive economic inequality in their classrooms, schools, neighborhoods, states, or countries? Which cues to they use to inform these perceptions (e.g., toys, clothing, appearance, speech, hobbies, housing)? When do these perceptions become accurate? |
| Socialization | What motivates parents, teachers, and other socialization agents to expose children to economic inequality, such as through cross-SES interactions? How do these adults discuss economic inequality with children, shaping children's own understanding of it? |
| Exposure | How does children's exposure to economic inequality change with age, such across school transitions? What are the differences between direct exposure (i.e., children's direct observations of economic inequality in their own immediate environments) and indirect exposure (i.e., children's observations of economic inequality in media, curricula, or conversations)? |
| Mechanisms | How do children interpret perceived economic inequality in their environment? For example, do they attribute it to individual traits (e.g., hard work, intelligence) or external structures (e.g., discrimination, exclusion)? When and why do they perceive inequality to be fair, just, and desirable? Do they infer from inequality an ethos of competition? |
| Consequences | How does perceived economic inequality affect children's mental health, physical health, moral development, personality development, social relationships, motivation, academic achievement, and political attitudes? When and why does perceived economic inequality have long-term effects that sustain even when children transition into more equal contexts? |
| Moderators | Do the effects of perceived economic inequality depend on whether children are from poor or affluent families? Do they depend on children's beliefs, such as their belief in meritocracy? |
| Intervention | What are the effects of creating or highlighting socioeconomic diversity in classrooms, schools, and neighborhoods? How do these effects differ between children from poor or affluent families? Can interventions ease the impact of perceived economic inequality (e.g., by offering children structural explanations for the inequality)? |

neighborhoods. Ironically, such efforts expose children to greater economic inequality. Without offering children a new way of understanding inequality, this could have unintended consequences, such as increased property crime among boys from low-SES backgrounds[11]. To better support children, we need evidence on how they perceive economic inequality. Such insights can inform programs that help children navigate it—by offering structural explanations for inequality, emphasizing the inherent value of socioeconomic diversity, and highlighting the unique strengths of individuals from diverse backgrounds[12].

Understanding economic inequality from children's perspective requires interdisciplinary collaboration. We need demographers, economists, and sociologists to map economic inequality across children's environments: their classrooms, schools, neighborhoods, states, and countries. We need psychologists, neuroscientists, and anthropologists to examine how actual inequality translates into perceived inequality, and how these perceptions shape children's thoughts, feelings, and behavior. We need epidemiologists, educational scientists, and political scientists specializing in large-scale panel research to track how these perceptions relate to children's long-term health, educational attainment, labor-market outcomes, and political attitudes. We need communication scientists to explore how perceptions of inequality are shaped through representation and discussions of economic inequality in schools, in traditional media, and on social media.

Establishing causality will be essential. Economic inequality covaries with many other contextual factors (e.g., poverty), so simply demonstrating an association between economic inequality and children's perceptions and outcomes will not suffice. Recent research offers creative strategies for establishing causality, even in dynamic real-world settings. For example, randomized controlled trials have tracked children as their families were randomly assigned to move from poor to more affluent neighborhoods, exposing them to economic inequality[11]. Similarly, natural experiments have followed children as they transitioned into schools with varying levels of economic inequality[3]. Extending this logic, randomized controlled trials can examine the causal real-world impact of offering children a new way of understanding inequality. Together, these approaches enable researchers to assess how children's exposure to and perceptions of inequality shape long-term development.

We call for interdisciplinary research to bring to light children's perspectives on economic inequality. Considering children's perspectives is essential not only for understanding how inequality shapes human development, but also for designing policies and programs that center children's lived experiences.

Eddie Brummelman [1] ✉, Richard E. Ahl [2], Elisabetta Aurino [3], Sahba N. Besharati [4], Benjamin W. Domingue [5], Catherine Lebel [6], Julia A. Leonard [7], Dana C. McCoy [8], Luca M. Pesando [9], Samuel S. Urlacher [10], David S. Yeager [11], Jason Yip [12] & Katherine McAuliffe [2]

[1]Research Institute of Child Development and Education, University of Amsterdam, Amsterdam, the Netherlands. [2]Department of Psychology and Neuroscience, Boston College, Chestnut Hill, MA, USA. [3]School of Economics, University of Barcelona, Barcelona, Spain. [4]Department of Psychology, University of the Witwatersrand, Johannesburg, South Africa. [5]Graduate School of Education, Stanford University, Stanford, CA, USA. [6]Department of Radiology, University of Calgary, Calgary, AB, Canada. [7]Department of Psychology, Yale University, New Haven, CT, USA. [8]Graduate School of Education, Harvard University, Cambridge, MA, USA. [9]Division of Social Science, New York University Abu Dhabi, Abu Dabi, United Arab Emirates. [10]Department of Anthropology, Baylor University, Waco, TX, USA. [11]Department of Psychology and Population Research Center, University of Texas at Austin, Austin, TX, USA. [12]The Information School, University of Washington, Seattle, WA, USA. ✉e-mail: e.brummelman@uva.nl

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

## Acknowledgements

This project was supported by seed funding from the Canadian Institute for Advanced Research (CIFAR) and the Jacobs Foundation to E.B., E.A., S.N.B., B.W.D., C.L., J.A.L., K.M., D.C.M., L.M.P., S.S.U., D.S.Y., and J.Y. During the writing, E.B. was supported by a Jacobs Foundation Research Fellowship (2020-1362-02) and an NWO Talent Programme Vidi grant (VI.Vidi.211.181), and D.S.Y. was supported by the National Science Foundation (2243530, 2201928, 2322330, and 2509858) and the Eunice Kennedy Shriver National Institute of Child Health and Human Development through the Population Research Center at The University of Texas at Austin (P2CHD042849). The funders had no role in the preparation of the manuscript or decision to publish.

## Author contributions

All authors contributed to the idea behind this Comment. E.B. played a lead role in writing the original manuscript. K.M. played a lead role in reviewing and editing the manuscript. R.E.A. played a lead role in literature search. E.B., R.E.A., E.A., S.N.B., B.W.D., C.L., J.A.L., D.C.M., L.M.P., S.S.U., D.S.Y., J.Y., and K.M. provided critical feedback and edits, and approved the final version of this manuscript.

## Competing interests

The authors declare no competing interests.

## Additional information

**Peer review information** The manuscript was considered suitable for publication without further review at Communications Psychology. Primary Handling Editor: Jennifer Bellingtier. A peer review file is available.

